# Borich's Needs Model Analysis of Smallholder Farmers' Competence in Irrigation Water Management: Case Study of Nkomazi Local Municipality, Mpumalanga Province in South Africa

**Mfanufikile Mabuza ***  and **Jorine T. Ndoro**

School of Agriculture, Faculty of Agriculture and Natural Sciences, University of Mpumalanga,
Cnr R40 and D725 Roads, Mbombela 1200, South Africa
* Correspondence: mabuza.mfanufikile247@gmail.com; Tel.: +27-65-644-6465

**Abstract:** Irrigated agriculture enables production intensification and crop diversification to improve food security. However, increasing irrigation water stress and inadequate competence of smallholder farmers in irrigation water management have the potential to exacerbate food insecurity. Therefore, this study seeks to determine smallholder farmers' competency needs in irrigation water management practices (IWMP). A convenience sampling method was employed to obtain a sample population of $n$ = 250. Descriptive statistics were employed to describe smallholder farmers' demographic characteristics. Borich's Needs Assessment Model (BNAM) was utilised to analyse smallholder farmers' competency needs. Results revealed that smallholder farmers perceived weed control (M = 4.90) and understanding the consequences of over- and under-irrigation (M = 4.48) as highly important practices. Results also revealed that smallholder farmers were only highly competent in weed control (M = 4.59). Moreover, results revealed that the top two most important competency needs for smallholder farmers are knowledge of drought-tolerant cultivars (MWDS = 6.83) and irrigation scheduling (MWDS = 5.05). From the survey findings, smallholder farmers' competency in IWMP is insufficient. It is recommended that the government, policymakers, and agricultural support services embark on sustainable agricultural development planning issues and develop a relevant training programme that is informed by smallholder farmers' competency needs.

**Keywords:** agricultural support services; extension; water management; Borich's needs model; training needs assessment; Nkomazi

## 1. Introduction

Agriculture is the main driver of food, employment opportunities, and household income in rural areas [1]. Agriculture has the potential to alleviate poverty while maintaining the ecosystem on which all people depend to improve living standards [2]. Smallholder farmers play a major economic role and support rural households by ensuring food availability. The second goal of the United Nations 2030 agenda on Sustainable Development Goals (SDGs) is to "end hunger, achieve food security and improve nutrition and promote sustainable agriculture" [3]. Smallholder farmers need to become competent in agronomic management practices in order to manage resources, improve performance and increase productivity. Ref. [4] defines competence as a series of desirable attributes that include abilities or capabilities such as knowledge of appropriate sorts, skills, problem solving, analysis, communication, and appropriate attitudes to satisfactorily perform a task. Competencies involve clusters of abilities, expertise, skills, and behaviours needed to succeed [5]. Improving the competence of smallholder farmers can promote sharing of knowledge with other farmers, thereby improving the skills of their workforce. Therefore, smallholder farmers need to be trained according to their hierarchy of competency needs to sustainably guarantee food security [6,7].

Smallholder farmers rely mainly on crop cultivation and animal husbandry as their main livelihood strategies [8]. Increasing water stress necessitates that smallholder farmers use irrigation water more competently to maximise yields [9]. Innovative knowledge and skills need to be disseminated to smallholder farmers to improve their competence in accomplishing sustainable agriculture. Improving sustainable agriculture depends on strong development and enhancing smallholder farmers' competency levels. Smallholder farmers' competence levels are the foremost means to mitigate the impacts of climate change on agricultural production. South Africa is a water-scarce country with an approximately 464 mm average annual rainfall [10], and irrigated agriculture is the largest user of water, estimated to be between 51% and 63% of the total water available [11]. Stress on irrigation water as a result of climate change has major production constraints for smallholder farmers. Thus, it is imperative to identify and assess smallholder farmers' competency needs on irrigation water management practices so as to draft a relevant training curriculum [5,12].

Relevant knowledge dissemination on irrigation water management practices has the potential to improve smallholder farmers' competence to enhance decision-making on irrigation water conservation. Ref. [13] indicated that an efficient and effective irrigation-based water use and management system is one of the main strategies for ending hunger in Africa by 2025. Incompetent smallholder farmers are greatly affected by climate change and face the challenge of coping with water scarcity, which exacerbates food insecurity, leading to hunger. Ref. [14] highlighted that irrigation water consumption has dropped from 80% to almost 50% in recent years. Ref. [15] further stated that South Africa's smallholder irrigation schemes are inefficient, unable to sustainably upsurge agricultural production and improve rural livelihoods. Irrigation schemes include multiple smallholder farmers' agricultural projects that rely on shared, decentralised systems to obtain irrigation water, and in some cases, from a shared water source [16]. Ref. [17] concluded that smallholder community irrigation schemes in Africa have proved to be highly unsustainable and face critical water management challenges. Smallholder farmers in irrigated agriculture experience more water losses owing to a lack of knowledge on irrigation scheduling, understanding of crop water requirement balance, utilisation of inappropriate irrigation methods, and how to operate and manage the systems [18–22].

Low literacy levels and the complexity of innovative irrigation systems have led smallholder farmers to continue using low-tech irrigation methods, resulting in inefficient management of irrigation water [16,21]. Low-tech irrigation methods involve the use of watering cans, buckets, and a hosepipe. According to [17], smallholder farmers' training in irrigation water management has been inadequate and not conducted with serious consideration. Similarly, Ref. [23] stated that smallholder farmers' competence level in irrigation water management practices is inadequate, making it challenging to achieve the 2030 SDGs. This is consistent with the findings of [24] that few agricultural advisory services provide training for smallholder farmers on irrigation water management. Scant literature exists on smallholder farmers' competency needs in irrigation water management practices in the Nkomazi local municipality, Mpumalanga Province in South Africa. Most researchers who studied South African smallholder farmers' training needs focused mainly on livestock training needs, land preparation training needs, pest and diseases management training needs, soil health and fertility management training needs, and horticulture training needs [25–29].

The South African Department of Agriculture has approved numerous agricultural support service organisations to help provide advice and train smallholder farmers on modern agricultural initiatives [30]. However, Ref. [17] stated that smallholder farmers lack competence in sustainable irrigation water management practices and adaptation of irrigation water conservation strategies. The inadequate information on smallholder farmers' competency needs on IWMP and increasing irrigation water scarcity led to this study to determine smallholder farmers' competency needs on irrigation water management practices in the Nkomazi local municipality. The findings are expected to provide modest but appropriate information that can give adequate insight to government, non-government

development stakeholders, and policymakers on strategies to initiate and enhance small-holder farmers' irrigation water management practices and production output; and ensure results that address the felt competency needs of smallholder farmers. Improving small-holder farmers' competence may help to achieve the SDGs, increase productivity, increase incomes and consumption, and enhance food security. The main objective of this study is to determine smallholder farmers' competency needs on 20 irrigation water management practices. The specific objectives of this study were to

I.   Determine smallholder farmers' perceived level of importance on 20 irrigation water management practices;
II.  Determine smallholder farmers' perceived level of competence on 20 irrigation water management practices;
III. Determine and rank smallholder farmers perceived competency needs on 20 irrigation water management practices.

## 2. Literature Review

### 2.1. Agricultural Support Services and Training Interventions

According to [31], the introduction of Non-Governmental Organisations (NGOs) laid a significant foundation for agricultural extension support services for smallholder farmers globally. Government and NGOs provide smallholder farmers with capital-enhancing inputs such as skill sets and knowledge to help them sustainably improve productivity, thus leading to the achievement of multiple SDGs. Nonetheless, according to the observations of [28], smallholder farmers receive the least training support. Ref. [28] further emphasises that farmers are trained or supported mainly on production procedures, neglecting farm management practices. Public extension services, in particular, face challenges such as having to facilitate land reforms, secure financial support and create sufficient initiatives aimed at developing smallholder farmers [32]. The provision of public extension services has failed to increase agricultural productivity and income [32]. It has been criticised for using outdated approaches that fail to meet smallholder farmers' needs in many developing countries, including South Africa [32]. Ref. [33] stated that for South Africa to achieve sustainable food security, agricultural support services must be well linked to all agricultural-related institutions and relevant research information, which is appropriately obtained from smallholder farmers' felt needs. Thus, agricultural support services must meet the competency/training needs of smallholder farmers so to disseminate relevant information, skills, and knowledge.

### 2.2. Smallholder Farmers Training Needs Assessment

Assessment of smallholder farmers' training needs is a step that is often missed in the process of developing training activities. According to [34], training needs assessment is the process of determining if there are training gaps and, if so, what knowledge and skills are needed to fill those discrepancies. It determines the levels of the current situation and whether smallholder farmers are proficient enough to achieve sustainable agriculture to end hunger. Training needs assessment is essential to select, interpret, describe, design, analyse, prioritise, and use the information to advance smallholder farmers' learning to improve production output [7]. It is the foundation of all training activities, which may include all agricultural management practices. Smallholder farmers' training needs result from underdeveloped skills, insufficient knowledge, or wrong work attitudes [7]. Assessing the perceived knowledge and skill gaps can help in identifying areas for improvement to overcome constraints hindering smallholder farmers from attaining sustainable agriculture. In Ref. [35], results revealed that farmers' professional competence is linked, for better or worse, with training needs.

Ref. [36] stated that smallholder farmers' training needs assessment has the potential to provide vibrant guiding principles regarding which skill and knowledge deficiencies need to be improved to upsurge yields while conserving production resources. An accurate training needs assessment can provide information to agricultural support programmes,

contributing to the overall training strategy and knowledge to be disseminated. The information may include urgent training areas to improve the competence of smallholder farmers. Smallholder farmers' training needs assessment and analysis must be carried out before training activities are structured since it underwrites the success of those activities.

Smallholder farmers face new water challenges owing to rapid population growth, pollution, climate change, and increased competition between water sectors [13]. Ref. [37] discussed that the shortage of irrigation water had become a restraint in achieving food security in all areas. Globally, it is estimated that agricultural food production and other agronomic products use approximately 70% of the freshwater withdrawn from rivers and groundwater [38]. Training on water accountability and water management practices for agriculture to meet these challenges remains a top priority as this can improve agricultural production and promote economic development [21,39,40]. Study findings by [24] revealed inadequate competence levels among smallholder farmers on irrigation water management practices. Moreover, Ref. [41] indicated that smallholder farmers in irrigation schemes face many agricultural production constraints to improve yields. The various crops they produce require different management skills. The study results by [23] revealed that the majority of smallholder farmers in South Africa have inadequate knowledge of sustainable agricultural productivity. Therefore, it is vital to determine competency needs assessment to understand knowledge and skills discrepancies among smallholder farmers' irrigation water management practices.

### 2.3. The Borich Needs Assessment Model

The BNAM is a research tool designed to enable the collection of data that can be weighted and hierarchically ranked in order of training needs [42]. Smallholder farmers are linked to a practical decision framework to improve training programmes by self-assessing educational gaps between their felt needs against the changing environment and the available modern agricultural information and technologies. The model measures different types of discrepancies [43] by comparing different levels of capabilities, such as smallholder farmers' perceived training needs, competency knowledge, ability to perform proficiently, and ability to produce sustainable competence. After assessing these proficiency dimensions, it becomes possible to analyse three types of discrepancies: knowledge discrepancies, gender discrepancies, and result discrepancies [43]. Ref. [44] opined that the model provides a strategy and a survey instrument that allows researchers to collect, weigh, and prioritise data hierarchically. Agricultural support services can amass relevant information that can help to draft a comprehensive set of learning objectives, content, materials, and methods. Smallholder farmers are associated with a practical decision-making framework, and determining training needs can be used to improve training programme curricula [44,45]. The model mainly uses a Likert-type scale questionnaire, which comprises a list of competencies ranking smallholder farmers' perceived capabilities and importance for each competency [43].

The BNAM was adopted because it best suits the holistic framework of this study. It has often been utilised productively by researchers in the assessment, analysis, and evaluation of training and competency needs [44,46–51]. Ref. [52] successfully used the BNAM to identify teaching competencies in the agricultural and life sciences faculty at the University of Florida. Ref. [42] excellently employed the BNAM to assess South Carolina teachers' in-service training needs of experienced agriculture teachers at the different stages of their teaching careers. Ref. [53] fruitfully utilised the BNAM in identifying the perceived level of importance and perceived competency levels of the in-service training needs of entry-level teachers. Ref. [53] used two sets of questionnaires; firstly, respondents were asked to rate each skill according to its perceived level of importance. The second questionnaire requested respondents to measure competence and express the need for each competency statement.

## 3. Materials and Methods

### 3.1. Study Area

This study was conducted in Nkomazi local municipality, geographically located in the east of the Ehlanzeni District between the north of Eswatini and the east of Mozambique. Two provincial highways, R570 and R571, connect it with Eswatini and Mozambique. The railway line and national highway (N4) form the Maputo Corridor [54]. The northern part is surrounded by the southeastern part of the Sabi River. According to [54], the city's geographic area covers approximately 478,754 ha, and according to [55], the municipal population is 410,907, of which 1.6% are white people and 97.7% are black people. In addition, 47.7% of the population are men, and 52.3% are women [54]. The main economic sectors in the area include agriculture, mining, and tourism, mostly originating from the towns of Komatipoort, Marloth, Kamhlushwa, and Malalane [55]. The climatic conditions in the municipality are generally temperate and warm, with much more rain in summer than in winter. The average mean annual rainfall for the municipality varies between almost 750 mm and 860 mm, with averages varying from 450 mm to 550 mm. The Nkomazi topographical structure is characterised by steep slopes and mountainous areas, mostly found in the western part and along the eastern margin of the municipality [54]. The Lebombo Plains are found in the vicinity of the Komati River and the Lebombo Mountains in the east of the municipality, characterised by flat to rolling landscapes. Flat areas are positioned in the central part between the Komati River and the mountainous western areas, becoming steeper in the south towards the Eswatini border [54].

The pre-eminently flat areas, loamy soils, the rainy seasons between October and March, and the perennial and dominant Komati and Crocodile Rivers promote agricultural practices within the area, where approximately 12,680 farmers practise subsistence agriculture [54]. According to [54], 75.3% of the municipal area is dominated by medium potential agricultural soils and only 15.3% by very low potential soils. In Nkomazi local municipality, smallholder farmers grow diverse crops, including vegetables, cotton, maize, grains, and sugarcane. Sugarcane farming is estimated to be practised by approximately 1243 smallholder growers across 37 irrigated farming projects [56,57], and there are about 1107 vegetable farmers. Figure 1 below shows the location of Nkomazi local municipality.

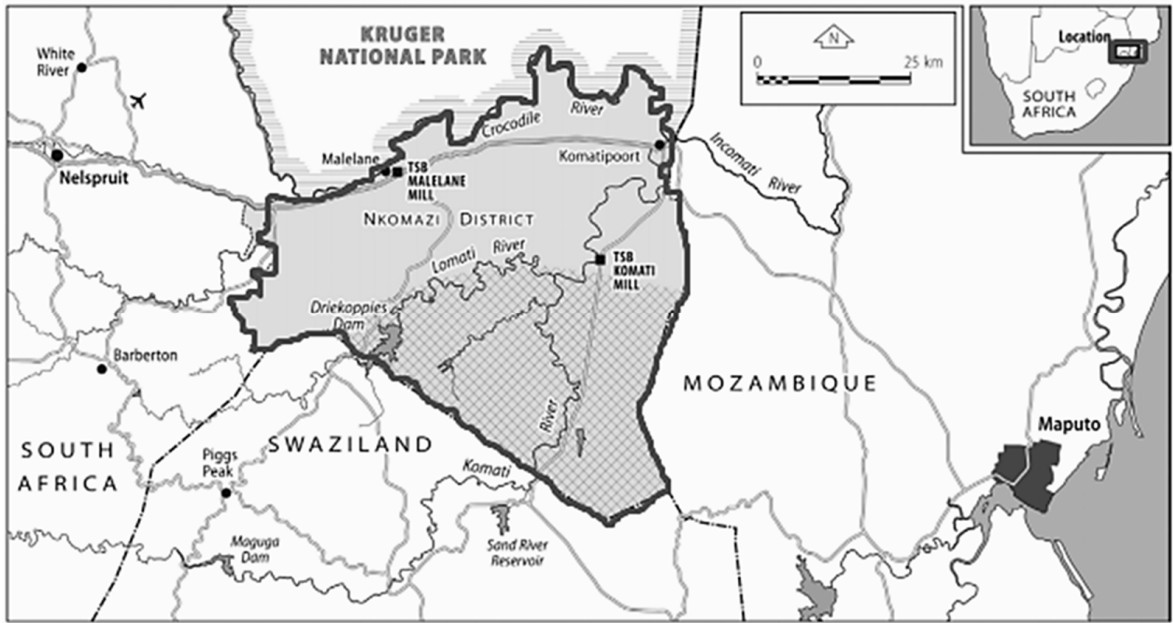

**Figure 1.** Nkomazi local municipality; Source [56].

### 3.2. Research Design and Target Population

A quantitative research design [58] was used in this study. Integrating the quantitative methods involves philosophical assumptions and theoretical frameworks [58]. Quantitative data involve data values in the form of counts or numbers, where each data set has a unique value associated with it [59]. The target population of this survey was smallholder farmers growing vegetables and sugarcane, as they utilise multiple irrigation systems and a large volume of water for irrigation. It is estimated that about 1243 smallholders practise sugarcane cultivation in 37 irrigated agricultural projects [56,57], and there are about 1107 vegetable farmers, making a total of 2350.

### 3.3. Sampling Method and Sampling Size

A convenience sampling [60] method was employed in the study area. This method relies on collecting data from the population willing to participate in a research study without any parameters or selection criteria [60]. The specific study communities for this research included Jepees, Schoemansdal, Langloope, Mzinti, Mbuzini, Skhwahlane, Stenbork, Tonga, Hoyi, Magogeni, and Naas, with an emphasis on smallholder farmers with irrigation schemes. Slovin's formula [61] was used to determine the sample sizes. The formula is applicable when estimating a population proportion and when the confidence coefficient is 95% [62]. The sample size used in this study was determined based on the cost of data collection and sufficient statistical power. A sample size with a confidence level of 95% and a margin error of 5% was utilised. A sample population of 250 smallholder farmers was used in this study.

### 3.4. Data Analysis

Structured questionnaire [63] was used for data collection. The questionnaire was divided into sections based on the research objectives, and contained smallholder farmers' demographic characteristics, 20 irrigation water management practices, and agricultural support training provisions on irrigation water management practices. Before the initial data collection process, the questionnaire was pre-tested to evaluate and improve the quality and effectiveness of primary research. To determine the reliability of the questionnaire, the mean of all possible split-half coefficients was measured using Cronbach's alpha ($\alpha$) from twenty smallholder farmers. The questionnaire showed an acceptable alpha coefficient, $\alpha = 0.856$ for importance and $\alpha = 0.879$ for competence. Tolerable alpha value estimates must range from 0.7 to 0.8; above 0.9 reflects an excellent consistency [64].

Smallholder farmers were asked to assess their perceived level of importance and perceived level of competence in 20 practices of irrigation water management. The 20 competency items were extracted from a range of literature, among others, as follows: training material for extension advisors in irrigation water management [22], irrigation practice and water management [65], Borich needs model analysis of extension agents' competence on climate-smart agricultural initiatives in South West Nigeria [66] and training needs; analysis of women in irrigation farming in the North West Province [26], sustainable water management in agriculture under climate change [67], participatory operation and maintenance of irrigation schemes [20,68], and the principles and practice of irrigation water management [69].

Smallholder farmers' perceived level of competence was measured on a five-point semantic differential Likert-type scale [70], ranging from 1 = very incompetent, 2 = least incompetent, 3 = undecided, 4 = competent, to 5 = highly competent. Smallholder farmers' assessment of perceived level of importance was also measured on a five-point semantic differential Likert-type scale with response options ranging from 1 = not important, 2 = least important, 3 = undecided, 4 = important, to 5 = highly important. Data were analysed on SPSS Version 27. Descriptive statistics were employed to describe smallholder farmers' demographic characteristics. Smallholder farmers' competency needs in irrigation water management were determined using Borich's Needs Assessment Model [43].

A discrepancy score (DS) [43,66] was calculated to obtain the difference between the importance rating and the competency rating of each competency. A weighted discrepancy score (WDS) [43] was calculated to assess and rank the competency needs of smallholder farmers. A mean weighted discrepancy score (MWDS) [43] was calculated to describe the overall ranking of each of the training areas. The competencies with the highest scores were those with the highest need and priority for training. The MWDS was calculated for each of the competencies using the following calculations:

- The difference between the importance rating and the competency rating of each competency of the irrigation water management practice was calculated for each respondent to generate the discrepancy score (DS) = importance rating minus the ability competence rating;
- The DS was then multiplied by the mean importance rating to generate the weighted discrepancy score (WDS) of each competency for the respondents;
- The sum of the weighted discrepancy scores divided by the number of observations was then used to compute the MWDS for each competency.

The model is shown as follows:

$$\text{MWDS} = \frac{(Iith - Cith) \times xI}{N}$$

where *I* = importance rating for each task, *C* = competency rating for each task, $\bar{x}$ mean of importance rating, and *N* = number of observations. The higher the MWDS, the greater the lack of smallholder farmers' competency in irrigation water management [29,71].

## 4. Results and Discussion

### 4.1. Smallholder Farmers' Education Level (n = 250)

Smallholder farmers' responses on education were categorised into seven groups, indicating the level of education of farmers from no school to tertiary level, as shown in Table 1. Results revealed that 62 participants had attained secondary education, accounting for 24.8% of the total participants. Of the total number of smallholder farmers who participated, 54 (21.6%) did not attend school and had no education at all. There were 48 (19.2%) participants who matriculated and 45 (18.0%) participants who had attained primary education. Table 1 further shows that 20 participants had achieved agricultural certificates, accounting for 8.0% of the total participants. There were 14 (5.6%) smallholder farmers who had a diploma, and only 7 participants had attained a degree, accounting for 2.8% of the total population. From Table 1, 64.4% represents a cumulative percentage that includes no school (21.6%) plus primary (18%) plus secondary (24.8%).

Table 1 reveals that smallholder farmers have a low level of education. This implies that there is a potential for inadequate competence in irrigation water management practices among smallholder farmers. Additionally, low levels of education may suggest that there is inefficient water use among smallholder farmers. Education level has a direct impact on smallholder farmers' decision-making process. The findings of this study concur with [72–74] that smallholder farmers have a low level of education. Among other socioeconomic factors, a low level of education inhibits the adoption of innovative agronomic practices. This has been demonstrated to seriously restrain the dissemination of innovative technologies for sustainable agriculture [74]. Technological and human capital advancements require a certain level of knowledge. Smallholder farmers' lack of education poses serious barriers to accessing useful institutions that disseminate information, knowledge, and skills. Thus, smallholder farmers struggle to meet quality standards set by fresh food markets and food processors [74,75].

**Table 1.** Smallholder farmers' demographic characteristics (*n* = 250).

|  |  | Frequency | Percent | Valid Percent | Cumulative Percent |
|---|---|---|---|---|---|
| Education level | No school | 54 | 21.6 | 21.6 | 21.6 |
|  | Primary | 45 | 18.0 | 18.0 | 39.6 |
|  | Secondary | 62 | 24.8 | 24.8 | 64.4 |
|  | Matriculated | 48 | 19.2 | 19.2 | 83.6 |
|  | Agriculture certificate | 20 | 8.0 | 8.0 | 91.6 |
|  | Diploma | 14 | 5.6 | 5.6 | 97.2 |
|  | Degree | 7 | 2.8 | 2.8 | 100.0 |
|  | Total | 250 | 100.0 | 100.0 |  |
| Irrigation methods | Flood | 4 | 1.6 | 1.6 | 1.6 |
|  | Sprinkler | 50 | 20.0 | 20.0 | 21.6 |
|  | Drip | 66 | 26.4 | 26.4 | 48.0 |
|  | Furrow | 89 | 35.6 | 35.6 | 83.6 |
|  | Centre pivot | 0 | 0 | 0 | 0 |
|  | Other | 41 | 16.4 | 16.4 | 100.0 |
|  | Total | 250 | 100.0 | 100.0 |  |

Competency needs assessment, appropriate training, and after-care training are the best mechanisms to improve smallholder farmers' competency levels. Enhancing smallholder farmers' education level is beneficial in improving their competence level. Education is an important human capital for smallholder farmers and society. Training and the acquisition of innovative knowledge, technologies, and skills promote socio-economic development. Smallholder farmers can acquire a comprehensive understanding of irrigation water management principles, introducing them to various innovative irrigation systems that can be selected, which gives them an understanding of the layout and operation of an irrigation system and how to set benchmarks for efficient irrigation water management on the farm [22]. Improving the level of education of smallholder farmers and societies is considered essential as it has the potential to advance smallholder farmers' decision making on irrigation water management.

*4.2. Smallholder Farmers' Irrigation Methods (n = 250)*

The results in Table 1 reveal that 89 smallholder farmers irrigate their crops using a furrow method, accounting for 35.6% of the total participants. Drip irrigation is the second most popular method, used by 66 smallholder farmers, accounting for 26.4% of the total number of participants. Moreover, results show that the sprinkler irrigation method is used by 50 smallholder farmers, who account for 20.0% of the total participants. Table 1 also reveals that 41 (16.4%) smallholder farmers use low-tech irrigation methods such as watering cans, buckets, and a hosepipe. None of the respondents use a centre-pivot system.

Most smallholder farmers use a furrow irrigation method. Table 1 results signify that, among other factors, the choice of an irrigation method used by smallholder farmers may be consistent with their education level. Smallholder farmers' competency level and the complexity of innovative irrigation systems impact the choice of an irrigation method. The study findings concur with [22,24,76] whose results reveal that most smallholder farmers use the furrow irrigation method. Short furrow irrigation is an indigenous irrigation system and is used predominantly by numerous smallholder farmers in South Africa [22]. This implies that cultural practices, beliefs, norms, and experience have an influence on the choice of an irrigation method. The observations of Refs. [37,77] revealed that indigenous knowledge is mostly held by the elderly and the uneducated. Furrow irrigation is where most irrigation water is lost because it is difficult to schedule, resulting in poor flow management of mainly surface runoff [21,22,24,78]. Smallholder farmers mainly use it because it is cheap and easy to use.

Further, this study found that the drip irrigation system is the second most used method. This signifies that smallholder farmers are gradually adopting innovative irriga-

tion methods, suggesting that continuous enhancement of smallholder farmers' competency level has the potential to improve the adoption of agronomic innovative practices. Among other constraints, adoption may be influenced by the cost of the systems and skills and knowledge levels. The findings of this study concur with the observation of [79] that smallholder farmers are gradually adopting the drip irrigation method, but are currently influenced by some factors, such as clogging of emitters, cost of pipelines, and shortage of water. In Ref. [57], survey results revealed that 48.8% of small-scale sugarcane farmers in the Nkomazi Municipality use the drip irrigation method—the best path to saving water and doubling irrigation productivity on irrigation schemes as a water-smart agricultural strategy [80,81].

Table 1 shows that 41 (16.4%) smallholder farmers use low-tech irrigation methods. The cumulative percentage indicates that 53.6% of smallholder farmers use low-tech irrigation methods. This may signify that smallholder farmers' competency level in irrigation and application efficiency is inadequate. The results of this study concur with [24,82], who observed that low-cost irrigation methods are widely used by smallholder farmers for vegetable production. Low-tech systems are difficult to schedule, resulting in over- and/or under-irrigation. Ref. [41] observed that smallholder farmers need great support as they lack efficient irrigation systems and scheduling instruments. Moreover, the issue of smallholder farmers' agricultural land ownership in South Africa may adversely affect the choice of an irrigation method. Lack of land ownership and insecure land rights may prevent smallholder farmers from making the necessary investments that would enhance agronomic management practices and economic value [83]. The results of the study by [84] revealed that only 7.6% of irrigated land in the North West Province in South Africa is privately owned, and 92.4% of the land is owned by the chief. The lack of smallholder farmers' access to agricultural land has limited their access to credit, which impacts smallholder farmers to invest in innovative agricultural initiatives. The study results of [85] revealed that Iran's agricultural production from 1981 to 2013 increased owing to improved farming practices and increased access to water through infrastructure and economic development. This implies that improving smallholder farmers' access to water, agricultural land, and improved infrastructure, training programs, and policies has the potential to upsurge agricultural production.

### 4.3. Smallholder Farmers' Perceived Level of Importance on 20 Competences (n = 250)

Smallholder farmers were asked to self-assess and rate their perceived level of importance on 20 irrigation water management practices using the following mean scale: not important (M = 1.0–1.49), Least important (M = 1.5–2.49), undecided (M = 2.5–3.49), important (M = 3.5–4.49), and highly important (M = 4.5–5.0). Table 2 shows that smallholder farmers perceived weed control (M = 4.90) and understanding the consequences of over- and under-irrigation (M = 4.48) as highly important competencies. Irrigation scheduling (M = 4.44), maintenance of irrigation system (M = 4.36), managing of irrigation system (M = 4.30), drought-tolerant cultivars (M = 4.27), soil, water, and plant relationships (M = 4.27), crop coefficient (M = 4.24), application efficiency (M = 4.19), irrigation efficiency (M = 4.16), soil moisture conservation techniques (M = 3.60), evaluation of irrigation systems (M = 3.56) and rainwater harvesting (M = 3.47) were perceived as important competencies for irrigation water management practices. Table 2 also reveals that smallholder farmers perceived overhead sprinkler irrigation (M = 2.46), the centre-pivot irrigation system (M = 2.08), and the micro-sprinkler irrigation system (M = 1.88) as the least important competencies. The rest of the practices were perceived as undecided.

**Table 2.** Smallholder farmers perceived competency needs on irrigation water management practices (n = 250).

| Irrigation Water Management Practices | Perceived Importance | | Perceived Competence | | Competency Needs | |
|---|---|---|---|---|---|---|
| | Mean | (SD) | Mean | (SD) | MWDS | Ranks |
| Drought tolerant cultivars | 4.27 | (0.96) | 2.67 | (1.38) | 6.83 | 1st |
| Irrigation scheduling | 4.44 | (0.73) | 3.29 | (1.26) | 5.05 | 2nd |
| Application efficiency | 4.19 | (0.98) | 3.04 | (1.19) | 4.83 | 3rd |
| Irrigation efficiency | 4.16 | (0.93) | 3.06 | (1.20) | 4.61 | 4th |
| Soil, water, and plant relationships | 4.27 | (0.83) | 3.19 | (1.27) | 4.60 | 5th |
| Evaluation of irrigation systems | 3.56 | (1.43) | 2.36 | (1.42) | 4.24 | 6th |
| Crop coefficient | 4.24 | (0.87) | 3.27 | (1.27) | 4.14 | 7th |
| Drip irrigation system | 3.42 | (1.82) | 2.24 | (1.47) | 4.02 | 8th |
| Calculations of on-farm water use efficiencies | 3.10 | (1.51) | 1.85 | (1.13) | 3.88 | 9th |
| Soil moisture conservation techniques | 3.60 | (1.32) | 2.62 | (1.37) | 3.53 | 10th |
| Understanding the consequences of over- and under-irrigation | 4.48 | (0.74) | 3.72 | (1.13) | 3.42 | 11th |
| Calibration of irrigation instruments | 2.94 | (1.51) | 1.80 | (1.07) | 3.35 | 12th |
| Maintenance of irrigation system | 4.36 | (0.94) | 3.63 | (1.12) | 3.17 | 13th |
| Rainwater harvesting | 3.47 | (1.69) | 2.60 | (1.43) | 3.01 | 14th |
| Managing of irrigation system | 4.30 | (0.92) | 3.61 | (1.09) | 2.99 | 15th |
| Irrigation operational costs | 3.13 | 1.46 | 2.17 | (1.28) | 2.99 | 16th |
| Weed control | 4.90 | (0.38) | 4.59 | (0.75) | 2.55 | 17th |
| Centre-pivot irrigation system | 2.08 | (1.49) | 1.34 | (0.83) | 1.55 | 18th |
| Overhead sprinkler irrigation | 2.46 | (1.50) | 1.88 | (1.16) | 1.45 | 19th |
| Micro sprinkler irrigation system | 1.88 | (1.20) | 1.55 | (0.93) | 0.62 | 20th |

Note: MWDS: mean weighted discrepancy scores.

Table 2 reveals that smallholder farmers attach high importance to weed control and understanding the consequences of over- and under-irrigation. This explains that smallholder farmers are aware of the great impact these initiatives have on producing good-quality crops. Numerous smallholder farmers control weeds manually. The results of this study are inconsistent with the findings of [86], whose results revealed that smallholder farmers perceived soil and water conservation, integrated pest management, and integrated disease management as very important practices. Moreover, the results concur with the findings of [5] that pre- and post-planting, women smallholder farmers perceived the appropriate application of herbicides and fungicides as one of the most important practices because weeds compete with crops for water, sunlight, and nutrients, and they harbour pests and diseases [87]. Ref. [88] found that water loss caused by weeds remains a major constraint to increased productivity and crop production worldwide.

In this study, smallholder farmers perceived 11 practices as important competencies for irrigation water management practices. These results may explain why smallholder farmers in the Nkomazi local municipality lack awareness and knowledge of the importance of irrigation water management. This is further explained by results shown in Table 1 which reveal that most 89 (35.6%) smallholder farmers use a furrow irrigation method. Furrow irrigation often leads to over-irrigation [78]. This ultimately explains the need to train smallholder farmers on irrigation scheduling, maintenance of irrigation systems, managing of irrigation systems, soil, water, and plant relationships, crop coefficient, application efficiency, irrigation efficiency, soil moisture conservation techniques, and evaluation of irrigation systems.

Additionally, in this study, smallholder farmers further perceive overhead sprinkler irrigation (M = 2.46), centre-pivot irrigation system (M = 2.08), and micro-sprinkler irrigation system (M = 1.88) as the least important competencies. This may reveal that these systems are complex for smallholder farmers to operate, and require smallholder farmers to be highly competent in operating and managing irrigation systems. In addition, smallholder farmers may consider these systems to be the least important because they

are expensive. Thus, the competency level, complexity, and cost of an irrigation system influence smallholder farmers' perceptions of the importance of irrigation methods.

### 4.4. Smallholder Farmers Perceived Level of Competence on 20 Competences (n = 250)

As shown in Table 2, a mean scale was used to determine smallholder farmers' perceived level of competence on 20 irrigation water management practices. Table 2 shows that smallholder farmers perceived weed control to be highly competent (M = 4.59), and they perceived the need for competence in understanding the consequences of over- and under-irrigation (M = 3.72), maintenance of irrigation system (M = 3.63), and managing of irrigation systems (M = 3.61). Smallholder farmers were undecided in eight competency areas and least competent in seven competency areas. However, they felt very incompetent on a centre-pivot irrigation system (M = 1.34).

The highly competent level of weed control may be explained by the reason that numerous smallholder farmers depend on manual weed control [89]. Manual weed control is a very effective and cheap method of weed control, although it is time-consuming. It does not require much extensive knowledge [89]. However, the results disagree with the findings of [26] that reveal that under pre- and post-planting, women smallholder farmers perceived the appropriate application of herbicides and fungicides as moderately important. This explains why smallholder farmers are not highly competent in chemical weed control. Chemical weed control requires more knowledge of the calibration of herbicides. Additionally, the allocated costs of herbicides influence smallholder farmers to control weeds manually.

The results of the study by [90] reveal that there is inadequate knowledge of the strategies to save water resources and sustain irrigation systems, which therefore concurs with the findings of this study that smallholder farmers do not have a very high level of competence in maintenance and management of irrigation systems. This implies that smallholder farmers face constraints such as frequent blockages/clogging of water emitters, which causes malfunctioning and additional maintenance costs, especially with drip irrigation systems.

The findings of this study disagree with [26], whose results reveal that female farmers engaged in irrigated agriculture perceived themselves to be very competent in irrigation scheduling and irrigation frequency. This may be explained by the results shown in Table 1, which show that most smallholder farmers use a furrow irrigation method. This suggests that lack of access to effective irrigation systems, such as drip systems, hinders smallholder farmers' competency in irrigation scheduling, resulting in more water loss. Similarly, Refs. [20–22] stated that smallholder farmers' irrigated agriculture experiences more water losses due to a lack of knowledge on irrigation scheduling, utilisation of inappropriate irrigation methods, and how to operate and manage the systems.

In this study, smallholder farmers perceived themselves to be least competent in micro-sprinkler irrigation systems (M = 1.55), calibration of irrigation instruments (M = 1.80), calculations of on-farm water use efficiencies (M = 1.85), overhead sprinkler irrigation (M = 1.88), irrigation operational cost (M = 2.17), drip irrigation system (M = 2.24), and evaluation of irrigation system (M = 2.36). These results may explain why few smallholder farmers use modern irrigation systems for water management, in that only 66 (26.4%) out of the total participants use the drip irrigation method. This implies that there is a potential for greater water loss among smallholder farmers due to a lack of knowledge, skills, and effective irrigation systems. Additionally, the results explain that smallholder farmers are highly competent in practices they perceive to be highly important. The results agree with the observations of [22] that agricultural extension officers provide minimal training to smallholder farmers on irrigation water management practices. This explains why smallholder farmers are least competent in key areas of irrigation water management practices. Smallholder farmers must be urgently trained in competency areas to advance their proficiency in irrigation water management practices.

### 4.5. Competency Needs of Smallholder Farmers on Irrigation Water Management Practices

Borich's procedure considers both the perceived knowledge of smallholder farmers and their perceptions of the importance of irrigation water management practices. Smallholder farmers' perceived importance and competence levels were then ranked using the calculated mean weighted discrepancy score (MWDS), as shown in Table 2. The higher the MWDS, the greater the competency needed [43] for smallholder farmers in irrigation water management practices. The MWDS shows the competency needs of smallholder farmers to manage irrigation water sustainably. Based on the MWDS ranking, results show that the urgent areas where smallholder farmers need more competencies were drought-tolerant cultivars (MWDS = 6.83), irrigation scheduling (MWDS = 5.05), application efficiency (MWDS = 4.83), irrigation efficiency (MWDS = 4.61), soil, water and plant relationships (MWDS = 4.60), evaluation of irrigation systems (MWDS = 4.24), crop coefficient (MWDS = 4.14), and the drip irrigation system (MWDS = 4.02). Competencies with the lowest MWDS were also identified, including the micro-sprinkler irrigation system (MWDS = 0.62), overhead sprinkler irrigation (MWDS = 1.45), and centre-pivot irrigation system (MWDS = 1.55).

The results shown in Table 2 reveal a need for more competence in practices that smallholder farmers perceived as important with an 'undecided' competency level. Moreover, the results explain that the smallholder farmers with the least competency level mostly perceived the most important practices with the highest priority for the need for training. The higher the priority, the more the farmers need knowledge and skills in the practice. This implies that smallholder farmers prioritised training on drought-tolerant cultivars followed by irrigation scheduling and application efficiency. The results are consistent with the findings of [91] that integrating drought, heat, and combined drought and heat tolerance cultivars with reference to maize varieties had clear advantages under current and future weather conditions. Drought-tolerant cultivars have the potential to increase production [91].

The results of this study differ from that of the findings of [26], which revealed that irrigation scheduling is the fourth in-service training area (fourth priority training need) for women in irrigation farming. This may explain that, among other factors, environmental aspects vary from region to region. The results presented in Table 1 show that most of the smallholder farmers use furrow and low-tech irrigation methods, and this is consistent with Table 2 that smallholder farmers need more competence in irrigation scheduling. In concurrence with the findings of [92], results revealed that smallholder farmers sought maximum training in integrated farming systems, integrated pest and disease management, and soil and water conservation technologies. Similarly, the results of this study are consistent with the findings of [86] that 20 (100%) farmers perceived training in irrigation water management as very important. This reveals that to achieve the objective of sustainable irrigation water management, training of smallholder farmers should be within their main priority competency areas, starting from the selection of drought resistance cultivars, irrigation scheduling, and application efficiency.

Innovative irrigation water management practices and agricultural extension services are expected to enhance smallholder farmers to better adapt to increasing irrigation water stress. However, smallholder farmers were not so competent in some of the related and vital irrigation water management practices. This results in inefficient water use by smallholder farmers. There is a need to improve agricultural extension training interventions in order to improve smallholder farmers' competency level in irrigation water management initiatives in the area. The human capital theory is centred on education and economic sectors, and asserts that the higher the education, the higher the economic return to society [26]. Smallholder farmers' competence enhancement would generally improve the process of sustainable agricultural expansion to improve the quality of life and the economic well-being of society while preserving resources. This emphasises the need to strengthen agricultural extension systems to train smallholder farmers through both conventional (i.e., demonstration fields, economic training, and organizational training)

and non-conventional (ICT, video, and mobile phone) methods [93] to enhance smallholder farmers' competence level. Strengthening innovative irrigation water management initiatives and human resources of extension agents to promote efficient water use among smallholder farmers is imperative. This has the potential to improve smallholder farmers' adoption decision-making process.

## 5. Conclusions

The study results reveal that the irrigation water management competency needs of smallholder farmers in the Nkomazi local municipality are generally high. Smallholder farmers self-evaluated their competence in most areas of irrigation water management practices as fairly low. Smallholder farmers' competency in irrigation water management practices is deficient. The importance of advancing smallholder farmers' competency in most irrigation water management practices is high. Most smallholder farmers use a furrow irrigation method, and this is where more water losses occur because it is difficult to schedule, resulting in surface runoff. The findings signify that there is a need to train smallholder farmers in most practices in irrigation water management so as to enhance their competency level and improve decision making. The competency needs are generally important and call for appropriate knowledge dissemination and demonstrations in most irrigation water management practices to improve irrigation water use efficiency. In all the 20 irrigation water management practices, a training curriculum has to prioritise training smallholder farmers on drought-tolerant cultivars, followed by knowledge about irrigation scheduling, application efficiency, irrigation efficiency, and knowledge of soil, water, and plant relationships.

It is recommended that the government, policymakers, and agricultural support services should embark on sustainable agricultural development planning issues and develop a relevant training programme that is informed by smallholder farmers' competency needs. Involving smallholder farmers in planning issues may help agricultural support services to disseminate information, knowledge, and skills that are relevant to their felt competency needs. Additionally, agricultural support services need to support smallholder farmers with effective irrigation systems for successful irrigation water management. The detailed examination of the smallholder farmers' training needs can help provide effective, relevant information that can give adequate insight to government and non-government development stakeholders on strategies to embark upon projects to enhance smallholder farmers' irrigation water management practices and production output, ensuring results that address the needs of smallholder farmers.

**Author Contributions:** Conceptualization, M.M. and J.T.N.; data collection and analysis, M.M.; M.M. wrote the first draft; supervision, J.T.N. All authors have read and agreed to the published version of the manuscript.

**Funding:** The research was funded by the Water Research Commission, South Africa; grant reference: C2020/2021-00222.

**Institutional Review Board Statement:** Ethics approval was granted by the University of Mpumalanga, Mbombela, South Africa.

**Informed Consent Statement:** Informed consent was obtained from all subjects involved in the study.

**Data Availability Statement:** All data have been included in the manuscript.

**Acknowledgments:** The authors would like to thank the Water Research Commission, South Africa, for funding the study. A special thanks to smallholder farmers from the Nkomazi local municipality for their participation in this study.

**Conflicts of Interest:** The authors declare no conflict of interest.

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
