# Peer review of "Borich’s Needs Model Analysis of Smallholder Farmers’ Competence in Irrigation Water Management: Case Study of Nkomazi Local Municipality, Mpumalanga Province in South Africa"

_sustainability, doi:10.3390/su15064935_

Round 1

Reviewer 1 Report

Abstract

An abstract must be meaningful and short; here author wrote too lengthy an abstract with unnecessary detail. Please focus on the basics of the Abstract, like providing the background of the study, aim and objectives, methodology, results, and recommendations.

Keywords must be different than the title of the manuscript; please use appropriate keywords to increase the visibility of your research.

Introduction

The introduction is well written, but please shorten this chapter, as it is too lengthy for the reader. Please focus on the key points and avoid repetition. The introduction must be short, to the point and describe previous study gaps. Here need to rewrite the aims and objectives, as the aims and objectives are too confusing; the authors want to determine the framer's perceived level of competency on 20 competencies of water management practices. I personally feel the author should remove competencies, and the sentence should be like this: To determine smallholder farmers perceived level of competence on 20 irrigation water management practices.

Please check grammar and spelling mistakes; English is too poor and needs professional assessment from any native English speaker.

Material and Methods,

Why did the authors use the smallholder farmers of Nkomazi Local Municipality? Is this area representative of the whole country's smallholder farmers, while in other areas, several environmental factors will differ and affect the recommendations of this study.

Why do authors use this Borich's Needs Assessment Model? The authors did not testify to the accuracy of this model. Please provide references in the Material and Method section; without appropriate citation, it seems like the authors newly develop these methods. If the authors developed any new method, please provide it in a supplementary file.

Result

Authors used self-assessed competency level of the smallholder farmers regarding agricultural practices on water management while most farmers were less educated or had a secondary diploma. Can authors rely on their provided information on agricultural practices like irrigation operational costs, rainwater harvesting, calibration of irrigation instruments, etc.? I feel that most farmers will be unaware of these practices, so their provided information raises serious questions about the accuracy of data used in this research. Please justify the research data used in this work.

Discussion

Discussion is poorly handled. Please write your findings and justify them by correlating them with previous findings. Please rewrite where it is needed. I am seriously concerned with the length of the discussion, as the discussion must be detailed, but sometimes too lengthy a discussion loses readers' interest.

Conclusion   

It lacks the future recommendations and overall importance of this research for the specific country worldwide. 

Author Response

Dear Reviewer 

I trust you doing good. Please see attached document for your attention.

Kind regards

Mabuza M.

Reviewer 2 Report

I would recommend a complete overhaul of the presented work to meet the standards of a peer-reviewed journal, both content-wise and in its presentation according to Sustainability's prescribed format.  

Author Response

(The authors gave the same response as above.)

Reviewer 3 Report

See "Comment to Authors"

Author Response

(The authors gave the same response as above.)

Reviewer 4 Report

1-     The author should have mentioned their innovation and its behind idea in the abstract and introduction section in more detail. the introduction needs some improvement to provide more information on similar works, the identified gaps, and the need for this study. The discussion section also needs some improvement regarding putting the findings of this study in the context of similar works. I suggest to see the "Iran's Agriculture in the Anthropocene".  

2-     The word "competency" is frequently used by the authors. But I don’t know what they mean by using this word?

3-     Lines 87-88, please describe such local irrigation method.

4-     Line 176, the unit is missing.

5-     Line 183, who did you use Slovin’s formula? Also line 204, detailed process of calculation done using Borich’s Needs Assessment Model is required to be mentioned in the text.

6-     Line 227 and table 1, It is suggested that classifications be done based on international classifications. For example, I cannot difference between two proposed categories such as " diploma" and "degree".

7-     Line 299, Since the interviewees have little education. Has the effect of daily conditions and temporary emotions been analyzed and sensitivity measured in the answers? For example, a series of same questions are asked from the same person at different times and the answers are compared with each other at different times

8-     How can the results of this scientific work be generalized in water resource development projects or similar experiences in the world? Can lifestyle and regional culture be effective in the level of reliability of these results?

9-     Can other important indicators such as the area of agricultural land or the number of human resources be effective on the results?

10- What are the types of education needed in the society under investigation? Only mentioning general points about the need for education cannot solve the obstacles in the region.

Author Response

(The authors gave the same response as above.)

Author Response

(The authors gave the same response as above.)

Reviewer 6 Report

Write the keyword small and specific. 

Author Response

(The authors gave the same response as above.)

Round 2

Reviewer 1 Report

I am pleased to accept the revised manuscript in present form. Thanks 

Author Response

Dear Reviewer

We thank you so much.

Kind regards

Mabuza M. and Ndoro J.

Reviewer 2 Report

No further comments. A thorough proof reading is recommended. 

Author Response

Dear Reviewer

We really appreciate your remarks to help us improve our research. Proof reading was done.

Kind regards

Mabuza M. and Jorine J.

Reviewer 4 Report

Thank 's for your kind attention. But the provided responses to questions number 7 and 9 cannot convince me.

Author Response

Dear Reviewer

We really appreciate your remarks to help us improve our research. All comments were addressed. Please see the attachment. 

Kind regards

Mabuza M. and Ndoro J.

Round 3

Reviewer 4 Report

thanks for your kind attention and provided respond letter. A scientific work should be able to be repeated. Based on the answer received regarding question 7, how can we expect that another researcher can repeat this work and reach the same results?

Author Response

Dear Reviewer

We really appreciate your remarks to help us improve our research. The comment was addressed.

Kind regards

Mabuza M.
